# Mitigating Mode Collapse By Sidestepping Catastrophic Forgetting

## Abstract

Generative Adversarial Networks (GANs) are a class of generative models used for various applications, but they have been known to suffer from the *mode collapse* problem, in which some modes of the target distribution are ignored by the generator. Investigative study using a new data generation procedure indicates that the mode collapse of the generator is driven by the discriminator's inability to maintain classification accuracy on previously seen samples, a phenomenon called Catastrophic Forgetting in continual learning. Motivated by this observation, we introduce a novel training procedure that dynamically spawns additional discriminators to remember previous modes of generation. On several datasets, we show that our training scheme can be plugged-in to existing GAN frameworks to mitigate mode collapse and improve standard metrics for GAN evaluation.

## 1 Introduction

Generative Adversarial Networks (GANs) (Goodfellow et al., 2014) are an extremely popular class of generative models that is not only used for text and image generation, but also in various fields of science and engineering, including biomedical imaging (Yi et al., 2019; Nie et al., 2018; Wolterink et al., 2017), autonomous driving (Hoffman et al., 2018; Zhang et al., 2018), and robotics (Rao et al., 2020; Bousmalis et al., 2018). However, GANs are widely known to be prone to *mode collapse*, which refers to a situation where the generator only samples a few modes of the real data, failing to faithfully capture other more complex or less frequent categories. While the mode collapse problem is often overlooked in text and image generation tasks, and even traded off for higher realism of individual samples (Karras et al., 2019; Brock et al., 2019), dropping infrequent classes may cause serious problems in real-world problems, in which the infrequent classes represent important anomalies. For example, a collapsed GAN can produce racial/gender biased images (Menon et al., 2020).

Moreover, mode collapse causes instability in optimization, which can damage not only the diversity but also the realism of individual samples of the final results. As an example, we visualized the training progression of the vanilla GAN (Goodfellow et al., 2014) for a simple bimodal distribution in the top row of Figure 1. At collapse, the discriminator conveniently assigns high realism to the region unoccupied by the generator, regardless of the true density of the real data. This produces a strong gradient for the generator to move its samples toward the dropped mode, swaying mode collapse to the opposite side. In particular, the discriminator loses its ability to detect fake samples it was previously able to, such as point **X•**. The oscillation continues without convergence.

From this observation, we hypothesize that the mode collapse problem in GAN training is closely related to Catastrophic Forgetting (McCloskey & Cohen, 1989; McClelland et al., 1995; Ratcliff, 1990) in continual learning. That is, since the distribution of the generated samples is not stationary, the discriminator *forgets* to classify the previously generated samples as fake, hindering convergence of the GAN minimax game. A promising line of works (Zhang et al., 2019b; Rajasegaran et al., 2019; Rusu et al., 2016; Fernando et al., 2017) tackle the problem in the supervised learning setting by instantiating multiple predictors, each of which takes charge in a particular subset of the whole distribution. Likewise, we also tackle the problem of mode collapse in GAN by tracking the severity of Catastrophic Forgetting by storing a few exemplar data during training, and dynamically spawning an additional discriminator if forgetting is detected, as shown in Figure 1. The key idea is that the added discriminator is left intact unless the generator recovers from mode dropping of that sample, essentially sidestepping catastrophic forgetting.

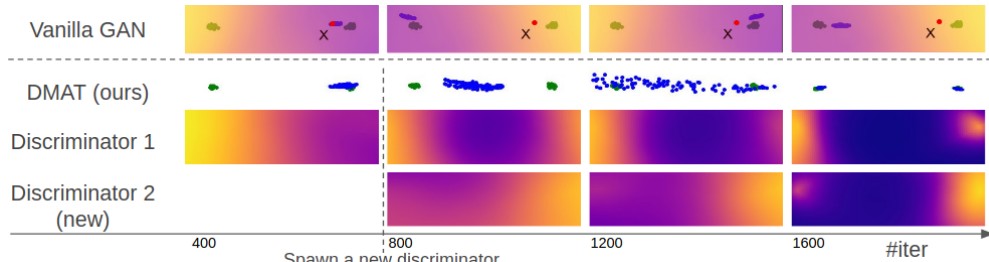

Figure 1: **Visualizing training trajectories**: We visualize the distribution of the real (green dots) and fake (blue dots) over the course of the vanilla GAN (top row) and our method (the second row and below). The background color indicates the prediction heatmap of the discriminator with blue being fake and warm yellow being real. Once the vanilla GAN falls into mode collapse (top row), it ends up oscillating between the two modes without convergence. Moreover, the discriminator's prediction at point X oscillates, indicating catastrophic forgetting in the discriminator. With our DMAT procedure, a new discriminator is dynamically spawned during training. The additional discriminator effectively learns the forgotten mode, guiding the GAN optimization toward convergence.

While the mode collapse problem has been tackled by many previous works, as discussed in Section 2, we show that our approach based on Catastrophic Forgetting can be added to any existing GAN frameworks, and is the most effective in preventing mode collapse. Furthermore, the improved stability of training boosts the standard metrics on popular GAN frameworks. To summarize, our contributions are:

- We propose a novel GAN framework, named Dynamic Multi Adversarial Training (DMAT), that prevents Catastrophic Forgetting in GANs by dynamically spawning additional discriminators during training.

- We propose a computationally efficient synthetic data generation procedure for studying mode collapse in GANs that allows visualizing high dimensional data using normalizing flows. We show that mode collapse occurs even in the recent robust GAN formulations.

- Our method can be plugged into any state-of-the-art GAN frameworks and still improve the quality and coverage of the generated samples.

## 2 RELATED WORKS

Previous works have focused on independently solving either catastrophic forgetting in supervised learning or mode collapse during GAN training. Among the efforts addressing mode collapse, a few prior works have proposed multi-adversarial solutions for mitigating mode collapse, similar to our work. In this section we review these works in detail and discuss our commonalities and differences.

### 2.1 MITIGATING MODE COLLAPSE IN GANS

Along with advancement in the perceptual quality of images generated by GAN (Miyato et al., 2018; Karras et al., 2019; Brock et al., 2018; Karras et al., 2020), a large number of papers (Durugkar et al., 2016; Metz et al., 2016; Arjovsky et al., 2017; Srivastava et al., 2017; Nguyen et al., 2017; Lin et al., 2018; Mescheder et al., 2018; Karras et al., 2019) identify the problem of mode collapse in GANs and aim to mitigate it. However, many of them do not attempt to directly address mode collapse, as it was seen as a secondary symptom that would be naturally solved as the stability of GAN optimization progresses (Arjovsky et al., 2017; Mescheder et al., 2018; Bau et al., 2019). While the magnitude of mode collapse is certainly mitigated with more stable optimization, we show that it is still not a solved problem. To explicitly address mode collapse, Unrolled GAN (Metz et al., 2016) proposes an unrolled optimization of the discriminator to optimally match the generator objective, thus preventing mode collapse. VEEGAN (Srivastava et al., 2017) utilizes the reconstruction loss on the latent space. PacGAN (Lin et al., 2018) feeds multiple samples of the same class to the discriminator when making the decisions about real/fake. In contrast, our approach differs in that our method can be plugged into existing state-of-the-art GAN frameworks to yield additional performance boost.

## 2.2 Multi-adversarial Approaches

The idea of employing more than one adversarial network in GANs to mitigate mode collapse or to improve the quality of generated images in general has been explored by several previous works. MGAN (Hoang et al., 2018) uses multiple generators, while D2GAN (Nguyen et al., 2017) uses two discriminators, and GMAN (Durugkar et al., 2016) and MicrobatchGAN (Mordido et al., 2020) possibly more than two discriminators that can be specified as a hyperparameter. On the other hand, based on our hypothesis on catastrophic forgetting, our method can *dynamically* add discriminators at training time, achieving superior performance than existing works.

## 2.3 Overcoming Catastrophic Forgetting in GAN

Catastrophic forgetting was first observed in connectionist networks by McCloskey & Cohen (1989). From then there has been a plethora of works to mitigate catastrophic forgetting in neural networks. These methods can be categorized into three groups: a) regularization based methods (Kirkpatrick et al., 2017) b) memory replay based methods (Rebuffi et al., 2017) c) network expansion based methods (Zhang et al., 2019a; Rajasegaran et al., 2019). Our work is closely related to the third category of methods, which dynamically adds more capacity to the network, when faced with novel tasks. This type of methods, adds *plasticity* to the network from new weights (fast-weights) while keeping the *stability* of the network by freezing the past-weights (slow-weights). Although our work incrementally adds more capacity to the network, we enforce stability by letting a discriminator to focus on a few set of classes, not by freezing its weights. The possibility of catastrophic forgetting of GANs has been discussed by Thanh-Tung & Tran (2020). However, the impact of catastrophic forgetting was mostly limited to theoretical analysis, and we found that the proposed solution deteriorates the performance on several GAN architectures such as BigGAN (Brock et al., 2019).

# 3 Proposed Method

In this section, we first describe our proposed data generation procedure that we use as a petri dish for studying mode collapse in GANs. The procedure uses random normalizing flows for simultaneously allowing training on complex high dimensional distributions yet being perfectly amenable to 2D visualizations. Next, we describe our proposed Dynamic Multi Adversarial Training (DMAT) algorithm that effectively detects catastrophic forgetting and spawns a new discriminator to prevent mode collapse.

## 3.1 Synthetic Data Generation with Normalizing flows

Mode dropping in GANs in the context of catastrophic forgetting of the discriminator is a difficult problem to investigate using real datasets. This is because the number of classes in the dataset cannot be easily increased, the classes of fake samples are often ambiguous, and the predictions of the discriminator cannot be easily visualized across the whole input space. In this regard, we present a simple yet powerful data synthesis procedure that can generate complex high dimensional multi-modal distributions, yet maintaining perfect visualization capabilities.

The proposed procedure begins by sampling from a simple two dimensional Gaussian distribution. The samples are then augmented with biases and subjected to an invertible normalizing flow (Karami et al., 2019) parameterized by well conditioned functions $g_i : \mathbb{R}^{d_i^0} \to \mathbb{R}^{d_i^1}$. Optionally, this function can be followed by a linear upsampling transformation parameterized by a $d_i^1 \times d_{i+1}^0$ dimensional matrix $A^i$ (Algorithm 1). The transformations are constructed to be analytically invertible thus allowing mapping the high dimensional output space to input (see Appendix D for more information). Note that the entire transform is deliberately constructed to be a bijective function so that every generated sample in $\hat{y} \in \mathbb{R}^D$ can be analytically mapped to $\mathbb{R}^2$, allowing perfect visualization on 2D space. Furthermore, by evaluating a dense grid of points in $\mathbb{R}^2$, we can also have a useful insight into discriminator's learned probability distribution on $\mathbf{z}$ manifold as a heatmap on a 2D plane.

This synthetic data generation procedure enables studying mode collapse in a controlled setting. This also gives practitioners the capability to train models on a chosen data complexity with clean two-dimensional visualizations of both the generated data and the discriminator's learnt distribution. This tool can be used for debugging new algorithms using insights from the visualizations. For

example, in the case of mode collapse, a quick visual inspection would give the details of which modes face mode collapse or get dropped from discriminator's learnt distribution.

## 3.2 DYNAMIC MULTI ADVERSARIAL TRAINING

---

**Algorithm 1** Synthetic Data Generation

---

**Input:** Mean $\{\mu_i\}_{i=1}^K$ and standard deviation $\{\sigma_i\}_{i=1}^K$ for initialization, $\{g_i\}_{i=1}^L$ well conditioned $\mathbb{R}^2 \rightarrow \mathbb{R}^2$ functions
Sample weights $\boldsymbol{w} \sim \text{Dirichlet}(K)$
/* Sample from 2D mixture of gaussians */
$\mathbf{x}_{2D} \sim \Sigma_{i=1}^N w_i \mathcal{N}(\mu_i, \sigma_i)$
$\mathbf{x}_{2D}^0 = \left[ [x_{2D}^0; 1], [x_{2D}^1; 1] \right]$
/* Randomly Initialized Normalizing Flow */
**for** $k = 1$ to $k = K$ **do**
  **if** k is even **then**
    $\boldsymbol{x}^k = \left[ \boldsymbol{x}_0^k, \boldsymbol{x}_1^k \cdot g_k(\boldsymbol{x}_0^k) \right]$
  **else**
    $\boldsymbol{x}^k = \left[ \boldsymbol{x}_0^k \cdot g_k(\boldsymbol{x}_1^k), \boldsymbol{x}_1^k \right]$
  **end if**
**end for**

---

**Algorithm 2** DSPAWN: Discriminator Spawning Subroutine

---

**Require:** Exemplar Data $\{e_i\}_{i=1}^m$
**Input:** Discriminator set $\mathbb{D} = \{f_w^i\}_{i=1}^K$
/* Check forgetting over exemplar images */
**for** $i = 1$ to $i = m$ **do**
  $\boldsymbol{s}[k] \leftarrow f_w^k(\boldsymbol{e}_i) \, \forall \, k \in \{1 \dots K\}$
  **if** $K * \max(\boldsymbol{s}) > \alpha_t * \sum_k \boldsymbol{s}[k]$ **then**
    Initialize $f_w^{K+1}$ with random weights $w$
    /* Spawn a new discriminator */
    Initialize random weight $w^{K+1}$
    $\mathbb{D} \leftarrow \{f_w^i\}_{i=1}^K \bigcup f_w^{K+1}$
    **break**
  **end if**
**end for**
**return** Discriminator Set $\mathbb{D}$

---

**Algorithm 3** D-MAT: Dynamic Multi-Adversarial Training

---

**Require:** $\boldsymbol{w}_0^i$, $\boldsymbol{\theta}_0$ initial discriminator & generator parameters, greediness parameter $\epsilon$, $\{T_k\}$ spawn warmup iteration schedule
$\mathbb{D} \leftarrow \{f_w^0\}$
**while** $\boldsymbol{\theta}$ has not converged **do**
  Sample $\{\boldsymbol{z}^{(i)}\}_{i=1}^B \sim p(z)$
  Sample $\{\boldsymbol{x}^{(i)}\}_{i=1}^B \sim \mathbb{P}_r$
  Sample $\{\sigma_1(i)\}_{i=1}^B \sim \text{Uniform}(1, K)$
  Sample $\{\alpha(i)\}_{i=1}^B \sim \text{Bernoulli}(\epsilon)$
  /* Loss weights over discriminators */
  Sample weights $\boldsymbol{m} \sim \text{Dirichlet}(K)$
  $\hat{\boldsymbol{x}}^{(i)} \leftarrow g_\theta(\boldsymbol{z}^{(i)})$
  $\sigma_2(i) \leftarrow \arg\min_k f_w^k(\hat{\boldsymbol{x}}^{(i)})$
  /* Discriminator responsible for $\hat{x}^{(i)}$ */
  $\sigma_z(i) \leftarrow \alpha(i)\sigma_1(i) + (1 - \alpha(i))\sigma_2(i)$
  /* Discriminator responsible for $x^{(i)}$ */
  $\sigma_x(i) \leftarrow \sigma_1(i)$
  /* Training Discriminators */
  $L_w \leftarrow \sum_{i=1}^B [f_w^{\sigma_x(i)}(\boldsymbol{x}_i) - 1]^- - [f_w^{\sigma_z(i)}(\hat{\boldsymbol{x}}_i) + 1]^+$
  **for** $k = 1$ to $k = |\mathbb{D}|$ **do**
    $w^k \leftarrow \text{ADAM}(L_w)$
  **end for**
  /* Training Generator */
  $s[k] \leftarrow \sum_{i=1}^B f_w^k(\hat{\boldsymbol{x}}^{(i)}) \, \forall \, k \in \{1 \dots |\mathbb{D}|\}$
  /* Weighed mean over discriminators */
  $L_\theta \leftarrow \text{sort}(\boldsymbol{m}) \cdot \text{sort}(s)$
  $\theta \leftarrow \text{ADAM}(L_\theta)$
  **if** more than $T_t$ warm-up iterations since the last spawn **then**
    $\mathbb{D} \leftarrow \text{DSPAWN}(\{f_w^i\})$
  **end if**
**end while**

---

Building upon the insight on relating catastrophic forgetting in discriminator to mode collapse in generator, we propose a multi adversarial generative adversarial network training procedure. The key intuition is that the interplay of catastrophic forgetting in the discriminator with the GAN minimax game, leads to an oscillation generator. Consequently, as the generator shifts to a new set of modes the discriminator forgets the learnt features on the previous modes. However if there are multiple discriminators available, each discriminator can implicitly *specialize* on a subset of modes. Thus even if the generator oscillates, each discriminator can remember their own set of modes, and they will not need to move to different set of modes. This way we can effectively *sidestep* catastrophic forgetting and ensure the networks do not face significant distribution shift. A detailed version of our proposed method is presented in Algorithm 3. **Spawning new discriminators**: We initialize the DMAT training Algorithm 3 with a regular GAN using just one discriminator. We also sample a few randomly chosen exemplar data points with a maximum of one real sample per mode, depending on dataset complexity. The exemplar data points are used to detect the presence of catastrophic forgetting in the currently active set of discriminators $\mathbb{D}$ and spawn a new discriminator if needed. Specifically

Table 1: ✓ indicates that the generator could effectively learn all the data modes, while ✗ means *despite best efforts with tuning* the training suffers from mode collapse (more than a quarter of the data modes are dropped). For each level, we show results with the SGD (left) & ADAM (right) optimizers. MNIST results with ADAM optimizer are provided for reference. We observe that MNIST is a relatively easy dataset, falling between `Level I` and `II` in terms of complexity.

| $g(\mathbf{z}) =$ | 1 | | $\mathbf{A}_{392\times2}$ | | $\mathbf{z}$ | | MLP | | MLP, $\mathbf{A}_{392\times2}$ | | MNIST |
|---|---|---|---|---|---|---|---|---|---|---|---|
| Label | `Level I` | | `Level II` | | `Level III` | | `Level IV` | | `Level V` | | - |
| GAN-NS (Goodfellow et al., 2014) | ✗ | ✓ | ✗ | ✓ | ✗ | ✗ | ✗ | ✗ | ✗ | ✗ | ✓ |
| WGAN (Arjovsky et al., 2017) | ✓ | ✓ | ✗ | ✓ | ✗ | ✓ | ✗ | ✗ | ✗ | ✗ | ✓ |
| Unrolled GAN (Metz et al., 2016) | ✓ | ✓ | ✓ | ✓ | ✓ | ✓ | ✓ | ✓ | ✗ | ✗ | ✓ |
| D2GAN (Nguyen et al., 2017) | ✓ | ✓ | ✓ | ✓ | ✓ | ✓ | ✓ | ✓ | ✗ | ✗ | ✓ |
| GAN-NS + DMAT | ✓ | ✓ | ✓ | ✓ | ✓ | ✓ | ✓ | ✓ | ✗ | ✗ | ✓ |

(Algorithm 2), we propose that if *any* discriminator among $\mathbb{D}$ has an unusually high score over an exemplar data point $\boldsymbol{e}_i$, this is because the mode corresponding to $\boldsymbol{e}_i$ has either very poor generated samples or has been entirely dropped. In such a situation, if training were to continue we risk catastrophic forgetting in the active set $\mathbb{D}$, if the generator oscillate to near $\boldsymbol{e}_i$. This is implemented by comparing the max score over at $\boldsymbol{e}_i$ to the average score over and spawning a new discriminator when the ratio exceeds $\alpha_t(>1)$. Further, we propose to have $\alpha_t(>1)$ a monotonically increasing function of $|\mathbb{D}|$, thus successively making it harder to spawn each new discriminator. Additionally, we use a warmup period $T_t$ after spawning each new discriminator from scratch to let the spawned discriminator train before starting the check over exemplar data-points.

**Multi-Discriminator Training:** We evaluate all discriminators in $\mathbb{D}$ on the fake samples but do not update all of them for all the samples. Instead, we use the discriminator scores to assign responsibility of each data point to only one discriminator. We use an $\epsilon$-greedy approach for fake samples where the discriminator with the lowest output score is assigned responsibility with a probability $1 - \epsilon$ and a random discriminator is chosen with probability $\epsilon$. In contrast, for real samples the responsible discriminator is always chosen uniformly randomly. In effect, we slightly prefer to assign the same discriminator to the fake datapoints from around the same mode to ensure that they do not forget the already learnt modes and switch to another mode. The random assignment of real points ensure that the same preferentially treated discriminator also gets updated on real samples. Further for optimization stability, we ensure that the real and fake sample loss incurred by each discriminator is roughly equal in each back-propagation step by dynamically reweighing them by the number of data points the discriminator is responsible for. We only update the discriminator on the losses of the samples they are responsible for. **Generator Training:** We take a weighted mean over the discriminators scores on the fake datapoints for calculating the generator loss. At each step, the weights each discriminator in $\mathbb{D}$ gets is in decreasing order of its score on the fake sample. Hence, the discriminator with the lowest score is given the most weight since it is the one that is currently specializing on the mode the fake sample is related to. In practice, we sample weights randomly from a Dirichlet distribution (and hence implicitly they sum to 1) and sort according to discriminator scores to achieve this. We choose soft weighing over hard binary weights because since the discriminators are updated in an $\epsilon$ greedy fashion, the discriminators other than the one with the best judgment on the fake sample might also hold useful information. Further, we choose the weights randomly rather than fixing a chosen set to ensure DMAT is more deadset agnostic since the number of discriminator used changes with the dataset complexity so does the number of weights needed. While a suitably chosen function for generating weights can work well on a particular dataset, we found random weights to work as well across different settings.

Table 2: **Quantitative Results on the Stacked MNIST dataset**: Applying our proposed dynamic multi adversarial training (DMAT) procedure to a simple DCGAN achieves perfect mode coverage, better than many existing methods for mode collapse.

| | GAN | UnrolledGAN | D2GAN | RegGAN | DCGAN | with DMAT |
|---|---|---|---|---|---|---|
| # Modes covered | $628.0 \pm 140.9$ | $817.4 \pm 37.9$ | $1000 \pm 0.0$ | $955.5 \pm 18.7$ | $849.6 \pm 62.7$ | $\mathbf{1000 \pm 0.0}$ |
| KL (samples $\parallel$ data) | $2.58 \pm 0.75$ | $1.43 \pm 0.12$ | $0.080 \pm 0.01$ | $0.64 \pm 0.05$ | $0.73 \pm 0.09$ | $0.078 \pm 0.01$ |

Table 3: **Quantitative Results on CIFAR10**: We benchmark DMAT against other multi-adversarial baselines as well as on several GAN architectures, observing consistent performance increase.

| Model | D2GAN | MicrobatchGAN | GAN-NS w/ ResNet | DMAT + GAN-NS | DCGAN | DMAT + DCGAN |
|---|---|---|---|---|---|---|
| IS | $7.15 \pm 0.07$ | 6.77 | $6.7 \pm 0.06$ | $\mathbf{8.1 \pm 0.04}$ | $6.03 \pm 0.05$ | $6.32 \pm 0.06$ |
| FID | - | - | 28.91 | **16.35** | 33.42 | 30.14 |
| Model | WGAN-GP w/ ResNet | DMAT + WGAN-GP | SN-GAN | DMAT + SN-GAN | BigGAN | DMAT + BigGAN |
| IS | $7.59 \pm 0.10$ | $\mathbf{7.80 \pm 0.07}$ | $8.22 \pm 0.05$ | $\mathbf{8.34 \pm 0.04}$ | 9.22 | $\mathbf{9.51 \pm 0.06}$ |
| FID | 19.2 | **17.2** | 14.21 | **13.8** | 8.94 | **6.11** |

## 4 RESULTS

We test our proposed method on several popular datasets, both synthetic and real & report a consistent increase in performance on popular GAN evaluation metrics such as Inception Score (Salimans et al., 2016) and Frechét Inception Distance (Heusel et al., 2017) with our proposed dynamic multi-adversarial training. Finally, we also showcase our performance in the GAN fine-tuning regime with samples on the CUB200 dataset (Welinder et al., 2010) which qualitatively are more colorful and diverse than an identical BigGAN finetuned without DMAT procedure (Figure 3).

### 4.1 SYNTHETIC DATA

We utilize the proposed synthetic data generation procedure with randomly initialized normalizing flows to visualize the training process of a simple DCGAN (Radford et al., 2015) in terms of the generated samples as well as discriminator's probability distribution over the input space. Figure 1 visualizes such a training process for a simple bimodal distribution. Observing the pattern of generated samples over the training iteration and the shifting discriminator landscape, we note a clear mode oscillation issue present in the generated samples driven by the shifting discriminator output distribution. Focusing on a single fixed real point in space at any of the modes, we see a clear oscillation in the discriminator output probabilities strongly indicating the presence of catastrophic forgetting in the discriminator network. Further such visualizations on more complex distributions (toy 8-D Gaussian rings) are added in the Appendix D.

**Effect of Data Complexity on Mode Collapse**: We use the flexibility in choosing transformations $g_i$ to generate datasets of various data distribution complexities as presented in Table 1. Choosing $g(z)$ with successively more complicated transformations can produce synthetic datasets of increasing complexity, the first five of which we roughly classify as Levels. The Levels are generated by using simple transforms such as identitym constant mapping, small Multi layer perceptrons and well conditioned linear transforms (**A**). On this benchmark, we investigate mode collapse across different optimizers such as SGD & ADAM (Kingma & Ba, 2014) on several popular GAN variants such as the non-saturating GAN Loss (GAN-NS) (Goodfellow et al., 2014), WGAN (Arjovsky et al., 2017) and also methods targeting mitigating mode collapse specifically such as Unrolled GAN (Metz et al., 2016) and D2GAN (Nguyen et al., 2017). We also show results of our proposed DMAT training procedure with a simple GAN-NS, which matches performance with other more complicated mode collapse specific GAN architectures, all of which are robust to mode collapse up to Level IV. The procedure can be extended to even more complicated distributions than Level V, but in practice we find all benchmarked methods to collapse at Level V. This indicates that in contrast to other simple datasets like MNIST (LeCun, 1998), Gaussian ring, or Stacked MNIST (Metz et al., 2016), the complexity of our synthetic dataset can be arbitrarily tuned up or down to gain insight into the training and debugging of GAN via visualizations.

### 4.2 STACKED MNIST

We also benchmark several models on the Stacked MNIST dataset following (Metz et al., 2016; Srivastava et al., 2017). Stacked MNIST is an extension of the popular MNIST dataset (LeCun et al., 1998) where each image is expanded in the channel dimension to $28 \times 28 \times 3$ by concatenating 3 single channel images from original MNIST dataset. Thus the resulting dataset has a 1000 overall modes. We measure the number of modes covered by the generator as the number of classes that are

Table 4: **BigGAN + DMAT Ablations on CIFAR10** (A) A relaxed spawning condition with small $\alpha$ and short warmup schedule that leads to large number of discriminators (>7) (B) Long warm-up schedules that spawn new discriminators late into training (C) A greedy strategy for assigning responsibility of fake samples ($\epsilon = 0$) (D) Flipping the data splitting logic with responsibilities of fake samples being random and of real being $\epsilon$-greedy and (E) Choosing the discriminator with lowest score for updating Generator instead of soft random weighting.

| Effect Ablation | Large $\|\mathbb{D}\|$ Small $\alpha$, Short $T_t$ | Spawn too late Long $T_t$ schedule | Greedy $\nabla$D $\epsilon = 0$ | Random for fake $\epsilon$-greedy for real | **1**-hot weight vector $\boldsymbol{m}$ | Proposed Method |
|---|---|---|---|---|---|---|
| IS | $8.83 \pm 0.04$ | $9.28 \pm 0.08$ | $9.31 \pm 0.06$ | $8.95 \pm 0.04$ | $9.25 \pm 0.05$ | $9.51 \pm 0.06$ |
| FID | $14.23$ | $9.37$ | $8.6$ | $12.5$ | $9.25$ | $6.11$ |

Table 5: **Per-class FID on CIFAR10**: FID improves consistently across all classes.

| Classes | Plane | Car | Bird | Cat | Deer | Dog | Frog | Horse | Ship | Truck | Avg |
|---|---|---|---|---|---|---|---|---|---|---|---|
| BigGAN | 24.23 | 12.32 | 24.85 | 21.21 | 12.81 | 22.74 | 17.95 | 13.16 | 12.11 | 18.39 | 8.94 |
| + DMAT | 20.50 | 10.30 | 23.48 | 18.48 | 11.51 | 19.41 | 11.50 | 12.24 | 10.69 | 12.94 | 6.11 |
| $\Delta\%$ | 18.2 | 19.6 | 5.8 | 14.8 | 11.3 | 17.2 | **56.1** | 7.5 | 11.7 | **42.1** | **46.3** |

generated at least once within a pool of $25,600$ sampled images. The class of the generated sample is identified with a pretrained MNIST classifier operating channel wise on the original stacked MNIST image. We also measure the KL divergence between the label distribution predicted by the MNIST classifier in the previous experiment and the expected data distribution.

**Understanding the forgetting-collapse interplay**: In Section 1, we discuss our motivation for studying catastrophic forgetting for mitigating mode collapse. We also design an investigative experiment to explicitly observe this interplay by comparing the number of modes the generator learns against the quality of features the discriminator learns throughout GAN training on the stacked MNIST dataset. We measure the number of modes captured by the generator through a pre-trained classification network trained in a supervised learning fashion and frozen throughout GAN training. To measure the amount of *'forgetting'* in discriminator, we extract features of real samples from the penultimate layer of the discriminator and train a small classifier on the real features for detecting real data mode. This implicitly indicates the quality and information contained in the the discriminator extracted features. However, the performance of classification network on top of discriminator features is confounded by the capacity of the classification network itself. Hence we do a control experiment, where we train the same classifier on features extracted from a randomly initialized discriminator, hence fixing a lower-bound to the classifier accuracy.

Referring to Figure 2, we observe a clear relation between the number of modes the generator covers at an iteration and the accuracy of the classification network trained on the discriminator features at the same iteration. In the vanilla single discriminator scenario, the classification accuracy drops significantly, indicating a direct degradation of the discriminative features which is followed by a complete collapse of G. In the collapse phase, the discriminator's learnt features are close to random with the classification accuracy being close to that of the control experiment. This indicates the presence of significant catastrophic forgetting in the the discriminator network.

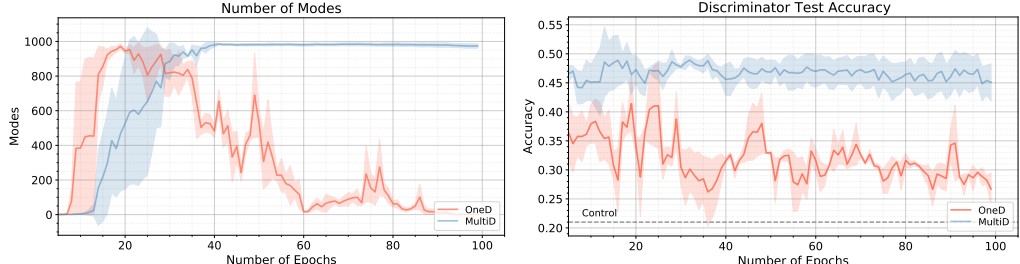

Figure 2: **Investigating the forgetting-collapse interplay:** We investigate our hypothesis that catastrophic forgetting is associated with mode collapse. To this end, on the left pane, we plot the magnitude of mode collapse by counting the number of modes produced by the generator. On the right pane, we assess the quality of the discriminator features by plotting the accuracy of linear classifier on top of the discriminator features at each epoch. In the original DCGAN model (*OneD*), the coverage of modes and the quality of discriminator features are both low and decreasing. In particular, the test accuracy from the discriminator's features drops almost to randomly initialized weights (shown as *control*). On the other hand, adding DMAT (*MultiD*) dramatically improves both mode coverage and the discriminator test accuracy.

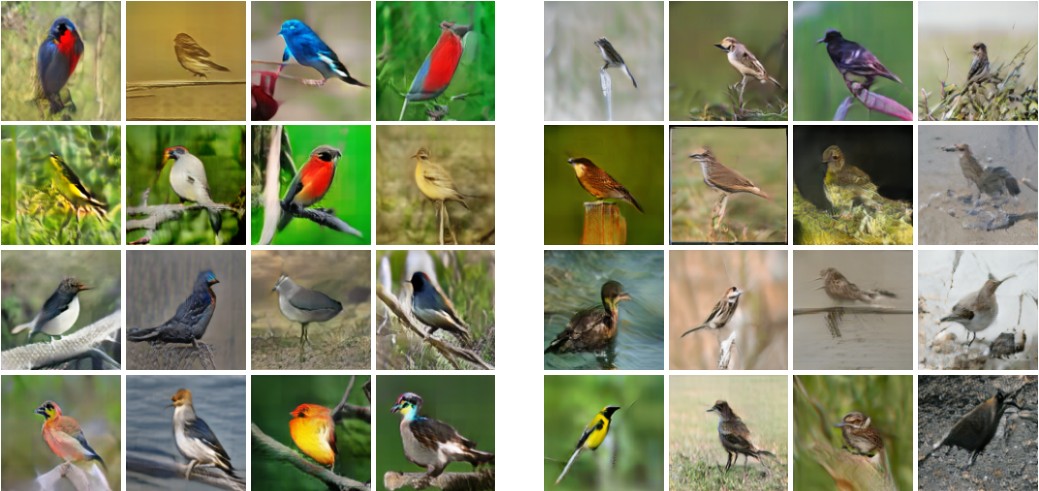

Figure 3: **Sample Diversity on CUB200:** We showcase samples from a BigGAN (pretrained on imagenet and finetuned on CUB200 with the DMAT procedure (left four columns) and from an identical BigGAN finetuned without DMAT (right four columns). We observe that while the sample quality is good for both setups, the samples generated with DMAT are more colorful & diverse, exhibiting bright reds and yellow against a variety of backgrounds. While the samples from vanilla fine-tuning are restricted to whites/grays & only a hint of color.

In contrast, training the same generator with the proposed DMAT procedure leads to stable training with almost all the modes being covered and the classification accuracy increasing before saturation. Catastrophic forgetting is thus *effectively sidestepped* by dynamic multi adversarial training which produces stable discriminative features throughout training that provide a consistent training signal to the generator thereby covering all the modes with little degradation.

### 4.3 CIFAR10

We extensively benchmark DMAT on several GAN variants including unconditional methods such as DCGAN (Radford et al., 2015), ResNet-WGAN-GP (Gulrajani et al., 2017; He et al., 2016) & SNGAN (Miyato et al., 2018) and also conditional models such as BigGAN (Brock et al., 2018). Table 3 shows the performance gain on standard GAN evaluation metrics such as Inception Score and Fréchet distance of several architectures when trained with DMAT procedure. The performance gains indicate effective curbing of catastrophic forgetting in the discriminator with multi adversarial training. We use the public evaluation code from SNGAN (Miyato et al., 2018) for evaluation. Despite having other components such as spectral normalization, diversity promoting loss functions, additional R1 losses & other stable training tricks that might affect catastrophic forgetting to different extents, we observe a consistent increase in performance across all models. Notably the ResNet GAN benefits greatly with DMAT despite a powerful backbone – with IS improving from 6.7 to 8.1, indicating that the mode oscillation problem is not mitigated by simply using a better model.

DMAT can also improve performance by over 35% even on a well performing baseline such as BigGAN (Table 3). We also investigate the classwise FID scores of a vanilla BigGAN and an identical BigGAN trained with DMAT on CIFAR10 and report the results in Table 5. Performance improves across all classes with previously poor performing classes such as 'Frog' & 'Truck' experiencing the most gains. Further, we also ablate several key components of the DMAT procedure on the BigGAN architecture with results reported in Table 4. We observe all elements to be critical to overall performance. Specifically, having a moderate $\alpha$ schedule to avoid adding too many discriminators is critical. Also, another viable design choice is to effectively flip the algorithm's logic and instead choose the fake points randomly while being $\epsilon$ greedy on the real points. We observe this strategy to perform well on simple datasets but lose performance with BigGAN on CIFAR10 (Table 4).

### 5 CONCLUSION

In summary, motivated from the observation of catastrophic forgetting in the discriminator, we propose a new GAN training framework that dynamically adds additional discriminators to prevent mode collapse. We show that our method can be added to existing GAN frameworks to prevent mode collapse, generate more diverse samples and improve FID & IS. As future work, we plan to apply our method to large scale experiments to prevent mode collapse in generating higher resolution images.

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
