# OpenReview forum: "Mitigating Mode Collapse by Sidestepping Catastrophic Forgetting"
_ICLR.cc/2021/Conference — Reject_

### Official Review · AnonReviewer3 · 2020-10-24
**The idea of the dynamic discriminator to address the mode collapse is new, but some missing related works and some experiments and discussion need to be justified**

**Rating:** 6
**Confidence:** 4

**Review:**

-- Summary --

This paper studies specifically the mode collapse problem of GAN. It hypothesizes that mode collapse in training GAN is closely related to catastrophic forgetting, and proposes a method to improve the catastrophic forgetting issue to mitigate the mode collapse. The proposed method uses multiple dynamically-spawned discriminators, in which additional discriminators are spawned to remember a few exemplars of data modes. It enables the model to avoid the oscillation in the discriminators' predictions, and handle well different parts of the data. The experiments demonstrate this multiple-spawned-discriminators scheme can be plugged to improves into various baseline GAN models on benchmark datasets (Stacked MNIST, CIFAR-10, CUB500 (qualitative only)). The paper also proposed a synthetic data generation procedure that can generate high-dimensional synthetic data that can be visualized at low-dimension for studying the mode collapse.

Overall, this paper is well-written and clear. Although the idea of multiple adversarial has been discussed in many existing works (some related works are missing in the paper, see below), this paper is different at posing a new scheme and specifically use it to address catastrophic forgetting. The experiments and study are sufficient to demonstrate the contributions.

However, the hypothesis of a relation between mode collapse and catastrophic forgetting, which was discussed in [c] (missing also), is not new. The algorithms are not too clear (notations, explanations) that need to be improved in the updated version. Additionally, some claims in the paper need to be justified.

[a] Multi-objective training of Generative Adversarial Networks with multiple discriminators.

[b] Generative Multi-Adversarial Networks.

[c] Self-Supervised GANs via Auxiliary Rotation Loss.

-- Strength --

S1 – The proposed algorithms of dynamic-spawned discriminators to address the catastrophic forgetting is new to me.

S2 – The experiments show significant improvements over baseline models.

S3 – The synthetic data generation procedure looks useful.

-- Weakness --

W1 – Some parts of the algorithms are not clear.

W2 – Quite missing to discuss some important works in the paper.

-- Details --

*Algorithms* - It would be a lot helpful and clearer if the authors explain the notations of Algorithms and relating them in the text discussions.

D1 – Algorithm 1 is a bit confusing with superscripts and subscripts of notations. Not sure superscripts or subscripts are the indices of samples or dimensions? According to the Algorithm, $x_{2D}^0$ is not used after being declared before the loop? And what is $N$ in Algorithm 1?

D2 – What are $[.]^-$ and $[.]^+$ of the loss function in Algorithm 3? Are the authors training GAN models with hinge losses but not discussed in the paper?

D3 – The random weights m used once to compute weighted means over the discriminators do not looks make sense to me. This random may have much affect to update the generator but their random values are chosen not from any prior pieces of information. I found some discussion about this in the section of “Generator Training”, yet can the author provide the ablation study to show the importance of using these random weights?

*Toy dataset*

D4 – In the study of Effect of Data Complexity on Mode Collapse (Table 1): The authors test the mode collapse by "more than a quarter of the data modes are dropped". How is "a quarter of the data modes" measured? Is there any threshold to decide a mode is dropped or not? And What is z for Level 3 of Table 1?

*Stacked MNIST experiment*

D5 – In the Stacked MNIST experiment, it's surprised that the original DCGAN can achieve quite a good mode collapse average if following the standard setup of Unrolled GAN (Metz et al., 2016). What is the architecture used for Stacked MNIST experiments?

*Forgetting-collapse interplay experiments*

D6 – In the proposed method, there are multiple discriminators. Are all features extracted from all discriminators chosen for the quality evaluation in forgetting-collapse interplay experiments?

D7 – The paper claimed: "we show ..., and is the *most effective* in preventing mode collapse". Although I agree that improving the catastrophic forgetting would improve the mode collapse, however, I would argue that there are likely other reasons causing mode collapse. For example, [d] shows the mode collapse still can happen in [c] due to the loss function of the classification. (Thanh-Tung et al., 2020) suggests the gradient penalty can address quite well the mode collapse. Can the authors compare the proposed method without gradient penalty compared to the same GAN models with only gradient penalty? Moreover, recent works [e,f,g] simply apply data augmentation for GAN training that significantly improves its performance. Therefore, I believe that data augmentation can also be helpful to address the mode collapse specifically in the case of limited data.

[d] Self-supervised GAN: Analysis and Improvement with Multi-class Minimax Game.

[e] Differentiable Augmentation for Data-Efficient GAN Training.

[f] On Data Augmentation for GAN Training.

[g] Training Generative Adversarial Networks with Limited Data.

---

### Official Review · AnonReviewer1 · 2020-10-25
**Review of 'Mitigating code collapse by sidestepping catastrophic forgetting'**

**Rating:** 7
**Confidence:** 3

**Review:**

The contributions of this paper revolve around the simple but interesting idea that mode collapse and missing modes in GANs are due (at least in part) by catastrophic forgetting in the discriminator, as the discriminator (and the generator) see a non-stationary training distributions due to their interaction. The authors provide toy experiments clearly illustrating this phenomenon and then propose an algorithm (DMAT) based on multiple discriminators taking charge for different parts of the input space (different modes). They show on several synthetic datasets with varying complexity that DMAT helps where other approaches fail, although DMAT (and all the other methods) still fail on the harder cases. They also show numerical improvements in standard GAN metrics against other methods (I don't know enough to be sure if these are SOTA, though). They also have nice experiments suggesting a strong link between mode collapse and catastrophic forgetting, thus also reinforcing the main claim.

Hence I see several good contributions around what seems to be a good idea. I only had a few minor comments, otherwise.

* there is generally a trade-off in generative models between missing modes and spurious modes; it would be good to make sure that we don't gain on one side by losing on the other.

* the algorithm is actually complicated and contains a lot of bells and whistles whose justification is not always clear (hence the statement in 4.1 that other methods are 'more complicated' seems unfair)

* the D2GAN performs as well as DMAT in table 2 but does not get the 'bold' treatment, this should be fixed.

* I do not find the difference between with and without DMAT in figure 3 compelling, so conclusion about this should also be toned down.

* several clarity issues:
  - alpha_t in Alg. 2 is not specified in the algorithm (although it is in the text)
  - the justification for alpha_t is not clear (and in particular the specifics of how it changes with t)
  - the justification for the experimental setup associated with figure 2 is not clear enough; why not keep the capacity of the classifier fixed (and not necessarily linear)ˆ?
  - 'with a maximum of one real sample per mode': how would you know, on real data, since the modes are not known a priori? hence this thinking only works for toy data where the modes are known.
  - English needs proofreading in many places

---

### Official Review · AnonReviewer4 · 2020-10-28
**Not bad results, but a somewhat unprincipled retread of a previously explored idea.**

**Rating:** 4
**Confidence:** 5

**Review:**

This paper addresses the well-known phenomenon of model collapse in generative adversarial networks (GANs). In particular, this paper identifies catastrophic forgetting of the discriminator as a potential source of mode collapse and proposes a multi-discriminator framework called DMAT as a solution. DMAT takes inspiration from expansion-based continual learning approaches, which add network capacity (in this case entire discriminators) to preserve past knowledge.

Pros:
1.	There are several excellent visualizations of generator oscillation behavior (Figure 1, Figure 4 <- I like the animations) that underpins DMAT’s design.
2.	Because DMAT is primarily based on adding discriminators to the model, it can be fairly simply combined with many GAN approaches. This is demonstrated for example in Table 3.
3.	Discriminators are dynamically added as needed, allowing the network to scale indefinitely to new modes. This also saves some computation at the beginning, when a large number of discriminators may not be necessary.
4.	A fairly wide range of experiments were run, and adding DMAT appears to lead to stronger results (higher Inception Score, lower Frechet Inception Distance) across a both an assortment of models and datasets.
5.	The quality of the writing is generally good.

Cons:
1.	I have concerns over the novelty of this work. The primary motivation of the paper (GAN mode collapse being due to catastrophic forgetting, leading to generator oscillations) is a known phenomenon and has been thoroughly explored before in [1,2]. In particular, [2] also takes a continual learning-inspired approach (regularization, as opposed to expansion) to propose an easy-to-add method, and visuals very similar to this submission’s Figure 4 can be found in both [1] and [2]. Given the strong degree of conceptual overlap, these works probably deserve more mention than a small reference to [1] in the Related Works. This paper presents the “GAN catastrophic forgetting” hypothesis in the Introduction as if it is the authors’ novel contribution. Additionally, multi-discriminator networks have also been well-explored in the past, with several examples mentioned in the Related Works. There are only limited comparisons with these methods in the Experiments, but Table 2 for example does demonstrate that D2GAN (which has 2 discriminators) also manages to cover all modes.
2.	The core methodology of DMAT is adding discriminators to capture new modes. While conceptually simple, this involves adding entire new networks, which can be expensive computationally. How many discriminators does DMAT typically add? In the continual learning literature, expansion of an entire network per task (equivalent to “per mode,” here) is considered very bad, and is the scenario that expansion strategies (some of which have sublinear parameter growth) specifically try to avoid.

3.	Discriminator assignment to samples and spawning/initialization is rather ad hoc, relying primarily on heuristics and intuition. A Dirichlet Process Mixture Model/Mixture-of-Experts approach such as in [3] (also a continual learning set-up, albeit for more traditional supervised classification) would be much more principled. Additionally, my understanding is that new discriminators are randomly initialized, which may be slow and inefficient.
4.	I don’t find the “fair comparison” in Appendix C to be particularly fair, as it’s unclear if the discriminator can effectively use the capacity of a wider network, and if it could, it’d likely disrupt the delicate minimax game between the discriminator and the generator; too strong of either a generator or discriminator severely disrupts GAN learning.
5.	I don’t fully understand the benefits of the proposed data generation process in Section 3.1. Settings like mixture of 8/25 2D Gaussians are fairly commonplace; being 2D means the discriminator decisions are easy to visualize, and the number of classes can be trivially increased/decreased by adding/removing classes. Using normalizing flows don’t make these synthetic distributions any more realistic either.

Miscellaneous:
1.	The title is a little too vague, and in my opinion “sidestepping” is somewhat inaccurate: despite some successful results on simple datasets, DMAT’s design doesn’t necessarily avoid catastrophic forgetting entirely. Given the plethora of continual learning strategies, the title should be more specific to DMAT’s strategy (namely model expansion by growing the number of discriminators).
2.	Alg 3 is a little confusing. The way it’s currently presented, it appears at first glance that a new discriminator is spawned on every iteration after $T_t$. It would be better if either the spawning logic from Alg 2 appeared in Alg 3, or the call to DSPAWN (Alg2) in Alg 3 made it clearer that a new discriminator wasn’t created on every iteration. Another thing worth noting: DSPAWN sounds very similar to “despawn,” which implies removing discriminators, not adding them.
3.	Both BigGAN and Side-tuning appear twice in the references.
4.	The gap between the Figure 3 caption and the main body text is a little too narrow.

Decision:
The experimental results presented in this paper are promising, but adding entire discriminator networks is computationally expensive, and I find DMAT’s formulation to be a little too unprincipled. Furthermore, while this specific combination may not have appeared in previous literature, discriminator catastrophic forgetting as a cause for mode collapse and multi-discriminator GANs have both been thoroughly explored before, leaving this submission with little novelty of its own. As a result, I recommend rejection.


[1] Hoang Thanh-Tung and Truyen Tran. On catastrophic forgetting and mode collapse in gans. arXiv preprint arXiv:1807.04015, 2020.

[2] Kevin J Liang, Chunyuan Li, Guoyin Wang, Lawrence Carin. Generative Adversarial Network Training is a Continual Learning Problem. arXiv preprint arXiv:1811.11083, 2018

[3] Soochan Lee, Junsoo Ha, Dongsu Zhang, Gunhee Kim. A Neural Dirichlet Process Mixture Model for Task-Free Continual Learning. ICLR 2020.


=========
Post-rebuttal
=========

I thank the authors for their thorough response, which is well argued. Overall, yes, I agree with the authors that there are some differences between their work and prior work. I do not claim that there is zero novelty here. My concern is whether there is enough for this venue, given the high degree of similarity. As the authors do acknowledge that the hypothesis of "catastrophic forgetting leads to GAN oscillations" was introduced in other work, then the primary contribution here is replacing one continual learning method for preventing catastrophic forgetting in GAN discriminators with another, and despite the rebuttal, I still find the proposed solution to be rather ad hoc. I keep my score.

---

### Official Review · AnonReviewer2 · 2020-11-05
**An interesting empirical study of mode collapse, evidence for a useful heuristic approach, but claims remain unsubstantiated.**

**Rating:** 5
**Confidence:** 3

**Review:**

The paper proposes and approach to mitigate mode collapse in arbitrary GANs. This is relies on the assumption that mode collapse relates to catastrophic forgetting in the GAN discriminator. The authors design a synthetic experiments based on a normalizing flow generator, allowing to quantify mode collapse. They propose a heuristic to avoid mode collapse of the discriminators by adding additional ones during training. The results show improvements with respect to baseline regarding mode covering and FID.

Strengths:
- the synthetic experiment to quantify mode collapse looks elegant,
- the authors report systemic improvement in FID over baselines.

Weaknesses:
- the authors fail to cite and compare their approach to another approach based on accumulating networks to improve GAN performance ("AdaGAN: Boosting Generative Models", Tolstikhin et al. 2017). There are also extra references therein.
- The relation between mode collapse and catastrophic forgetting in not clearly supported. This can be seen in Fig. 2, were despite the authors statements, there is no clear covariation between forgetting and collapse.
- While the systematic improvement of mode covering provided by the method is not surprising, the systematic FID improvement are weakened by the absence of detailed (e.g. supplemental) material to reproduce the results and make sure this result is solid.
- The authors make a link between mode collapse and oscillations, although the later is now a well studied and general phenomenon with solid theoretical results, that can be dealt with by modifying the optimization algorithm (Daskalakis et al, Training GANs with optimism).

Overall, this seems and interesting research direction, but evidence for the main claim of the paper is lacking. In the current state I tend to recommend rejection.


Perhaps the authors could find investigate more in depth going through all their experiments, whether their is a clear causal link between mode collapse and catastrophic forgetting. This is likely tricky, in the sense that mode collapse will likely lead to a loss of discriminator performance. Also the authors could compare their approach with alternative solutions, that do not rely on this collapse-forgetting link, such as the two papers cited above.

---

### Public Comment · ~Thang_Doan1 · 2020-11-15
**Related works**

 Hi,

This is a really interesting idea to add additional discriminators dynamically (I once thought about it too).
I believe [1,2] are relevant items for your "multiple discriminators" related work section.

[1][On-Line Adaptative Curriculum Learning for GANs](https://ojs.aaai.org//index.php/AAAI/article/view/4224)
[2][Multi-objective training of Generative Adversarial Networks with multiple discriminators](http://proceedings.mlr.press/v97/albuquerque19a.html)

---

### Decision · Program_Chairs · 2021-01-07
**Final Decision**

**Decision:**

Reject

**Comment:**

The authors propose an approach to mitigate mode collapse phenomena in GANs. Motivated by the intuition that mode collapse stems from catastrophic forgetting of the discriminator, the authors propose a solution inspired by recent research in continual learning and dynamically add new discriminators during training. The authors empirically demonstrate that combining the proposed scheme with existing GANs leads to improvements in terms of Inception Score and FID.

This paper is trying to address a significant problem for the generative modeling community. The reviewers appreciated the clarity of writing, the empirical results, and the idea of using normalising flows for an elegant visualisation. However, the reviewers have pointed out several major issues which were not adequately addressed by the authors. The first one is the clear failure to position the work with respect to related work. In fact, the main idea related to catastrophic forgetting was already established in [1,2] and subsequent works. Secondly, the improvements over the baseline come at a significant computational overhead which is extremely challenging and impractical. Finally, given the trend that large-scale models achieve significantly better results in practice, the proposed approach is not only impractical, but potentially extremely wasteful. Given very limited novelty, failure to position the work, and impracticality of the proposed solution, I will recommend rejection.

[1] https://arxiv.org/abs/1810.11598

[2] https://arxiv.org/abs/1911.06997